# Variation in Nutritional Components and Antioxidant Capacity of Different Cultivars and Organs of *Basella alba*

**DOI:** 10.3390/plants13060892

**Published:** 2024-03-20

**Authors:** Yi Zhang, Wenjuan Cheng, Hongmei Di, Shihan Yang, Yuxiao Tian, Yuantao Tong, Huanhuan Huang, Victor Hugo Escalona, Yi Tang, Huanxiu Li, Fen Zhang, Bo Sun, Zhi Huang

**Affiliations:** 1College of Horticulture, Sichuan Agricultural University, Chengdu 611130, China; 202001780@stu.sicau.edu.cn (Y.Z.); 2021205035@stu.sicau.edu.cn (H.D.); 202101726@stu.sicau.edu.cn (S.Y.); S20162807@stu.edu.cn (Y.T.); 20134983@stu.edu.cn (Y.T.); hh820423@163.com (H.H.); 13920@sicau.edu.cn (Y.T.); 10650@sicau.edu.cn (H.L.); zhangf@sicau.edu.cn (F.Z.); 2Institute of Agricultural Resources and Environment, Tianjin Academy of Agricultural Sciences, Tianjin 300384, China; tjnkyxmk@tj.gov.cn; 3The State Key Laboratory of Vegetable Biobreeding, Tianjin Academy of Agriculture Sciences, Tianjin 300192, China; 4Faculty of Agricultural Sciences, University of Chile, Santa Rosa 11315, Santiago 8820808, Metropolitan Region, Chile; vescalona@uchile.cl

**Keywords:** *Basella alba*, nutritional components, total proanthocyanidins, total phenolics, ABTS, FRAP, antioxidants, cultivars, organs

## Abstract

*Basella alba* is a frequently consumed leafy vegetable. However, research on its nutritional components is limited. This study aimed to explore the variation in the nutritional components and antioxidant capacity of different cultivars and organs of *Basella alba*. Here, we primarily chose classical spectrophotometry and high-performance liquid chromatography (HPLC) to characterize the variation in nutritional components and antioxidant capacity among different organs (inflorescences, green fruits, black fruits, leaves, and stems) of eight typical cultivars of *Basella alba*. The determination indices (and methods) included the total soluble sugar (anthrone colorimetry), total soluble protein (the Bradford method), total chlorophyll (the ethanol-extracting method), total carotenoids (the ethanol-extracting method), total ascorbic acid (the HPLC method), total proanthocyanidins (the p-dimethylaminocinnamaldehyde method), total flavonoids (AlCl_3_ colorimetry), total phenolics (the Folin method), and antioxidant capacity (the FRAP and ABTS methods). The results indicated that M5 and M6 exhibited advantages in their nutrient contents and antioxidant capacities. Additionally, the inflorescences demonstrated the highest total ascorbic acid and total phenolic contents, while the green and black fruits exhibited relatively high levels of total proanthocyanidins and antioxidant capacity. In a comparison between the green and black fruits, the green fruits showed higher levels of total chlorophyll (0.77–1.85 mg g^−1^ DW), total proanthocyanidins (0.62–2.34 mg g^−1^ DW), total phenolics (15.28–27.35 mg g^−1^ DW), and ABTS (43.39–59.16%), while the black fruits exhibited higher levels of total soluble protein (65.45–89.48 mg g^−1^ DW) and total soluble sugar (56.40–207.62 mg g^−1^ DW) in most cultivars. Chlorophyll, carotenoids, and flavonoids were predominantly found in the leaves of most cultivars, whereas the total soluble sugar contents were highest in the stems of most cultivars. Overall, our findings underscore the significant influence of the cultivars on the nutritional composition of *Basella alba*. Moreover, we observed notable variations in the nutrient contents among the different organs of the eight cultivars, and proanthocyanidins may contribute significantly to the antioxidant activity of the fruits. On the whole, this study provides a theoretical basis for the genetic breeding of *Basella alba* and dietary nutrition and serves as a reference for the comprehensive utilization of this vegetable.

## 1. Introduction

*Basella alba*, commonly referred to as Malabar spinach or Ceylon spinach, is an annual herbaceous plant classified under the family *Basellaceae* and the order Caryophyllales [1]. It originated in tropical Asia; it is a widely consumed green leafy vegetable known for its smooth, fleshy appearance; and it enjoys popularity in regions with tropical and temperate climates. The main edible parts of *Basella alba* are its thick, semi-succulent, mucilaginous leaves and its tender stems. *Basella alba* is commonly incorporated into soups to mitigate its sticky texture, and it serves as an excellent addition to salads and dips [2,3].

*Basella alba* possesses considerable economic, nutritional, and medicinal value, representing a valuable source of income for marginal and small farmers residing near major urban centers [2]. In addition, it serves as a promising source of safe and effective skin-moisturizing polysaccharides, with the potential for industrial-scale adaptation [4]. Previous studies have demonstrated that *Basella alba* is rich in nutritional components such as carbohydrates, protein, and ascorbic acid [2,5,6,7]. This plant garners consumer attention as a leafy vegetable with both nutritional value and healthcare benefits [8]. It is commonly employed in traditional Indian and Chinese medicine due to its medicinal properties, including the treatment of constipation and diuretic, detoxifying, and anti-inflammatory effects [3]. The vines of *Basella alba* produce a substantial quantity of fleshy, stalkless, ovoid or spherical dark blue fruits that turn purple when they mature. Unfortunately, these fruits are typically discarded by farmers [9,10].

Numerous studies have shown that antioxidants are crucial for maintaining human health and preventing diseases, as they reduce oxidative stress [11,12]. Ascorbic acid, proanthocyanidins, flavonoids, and phenolics are important antioxidants with potential anti-allergic, anti-inflammatory, and antibacterial properties [13,14]. Therefore, measuring the antioxidant activity of plants is essential to ensure the quality of functional plants and to study the effectiveness of plant antioxidants in preventing and treating diseases associated with oxidative stress.

One of the main factors that influences the types and concentrations of the nutritional components in this leafy vegetable is the type of cultivar [15,16]. Prior studies have identified variations in mineral elements and phytochemical contents among different cultivars of lettuce, *Perilla frutescens* (L.) Britt., and cauliflower [17,18,19]. Additionally, different plant organs exhibit variations in nutritional component contents. For instance, Sun et al. found that the lateral bud of baby mustard (*Brassica juncea* var. *gemmifera*) showed greater concentrations of health-promoting compounds than the swollen stem [20]. Broccoli florets possess the highest protein content, and stems exhibit a high crude fiber content, whereas leaves and seeds demonstrate notable antioxidant abilities [21,22,23,24,25].

Several studies have explored the nutritional components and health functions of *Basella alba*, but many have been limited to the analysis of individual organs [3,26]. This study utilized eight typical cultivars (M1–M8) of *Basella alba* to investigate the variations in nutritional components and antioxidant capacity across inflorescences, green fruits, black fruits, leaves, and stems (Figure 1). The aim was to comprehensively understand the utilization value of *Basella alba*. Therefore, this study provides a theoretical basis for the genetic breeding of *Basella alba*, which is essential for improving its dietary nutrition value for people. Additionally, this study explored the comprehensive utilization of abandoned organs in the production practice of *Basella alba*.

## 2. Results

### 2.1. Total Soluble Sugar

The cultivar with the highest total soluble sugar content was M5 (173.64 mg g^−1^ DW), while M2 (104.83 mg g^−1^ DW) exhibited the lowest content. The stems, except for those of M3, had the highest total soluble sugar contents, which were significantly higher than those of the other organs, followed by the leaves, inflorescences, black fruits, and green fruits. The difference between the stems and green fruits in the eight cultivars varied from 3.7 to 7.6 times. In most cultivars, the black fruits contained significantly more total soluble sugar than the green fruits (1.1 to 3.0 times more) (Figure 2A).

### 2.2. Total Soluble Protein

The total soluble protein content in M5 (92.74 mg g^−1^ DW) was significantly higher than those in the other cultivars. Furthermore, it was 1.6 times higher than that in the cultivar with the lowest content (M3, 58.44 mg g^−1^ DW). The organ with the highest total soluble protein content varied among cultivars. The highest content was found in the inflorescences in M1 (72.88 mg g^−1^ DW) and M7 (110.73 mg g^−1^ DW), black fruits in M2 (71.90 mg g^−1^ DW) and M4 (82.89 mg g^−1^ DW), and leaves in M5 (120.08 mg g^−1^ DW). However, the lowest total soluble protein content was found in the stems in all cultivars, ranging from 45.82 to 72.92 mg g^−1^ DW. Moreover, the black fruits contained significantly more total soluble protein than the green fruits in M2, M3, M4, M5, and M7; the opposite was observed in M1 (Figure 2B).

### 2.3. Total Chlorophyll and Total Carotenoids

M5 exhibited a significantly higher total chlorophyll content than the other cultivars (2.81 mg g^−1^ DW), while the cultivar M7 had the lowest total chlorophyll content (2.24 mg g^−1^ DW). The total chlorophyll contents were highest in the leaves, followed by the stems, green fruits, and black fruits (except in M2 and M3), while the inflorescences (except in M5 and M7) had the lowest contents. Meanwhile, the total chlorophyll contents in the leaves ranged from 6.28 mg g^−1^ DW (M8) to 8.68 mg g^−1^ DW (M5), and in the inflorescences, they ranged from 0.56 mg g^−1^ DW (M3) to 1.29 mg g^−1^ DW (M7), with the largest difference between the leaves and inflorescences being 13.8-fold in M2 (Figure 3A).

The cultivar M2 (0.23 mg g^−1^ DW) showed a significantly higher total carotenoid content compared to the others, and M4 had the lowest. The total carotenoid content in M2 was 1.3 times higher than that in M4. Moreover, the total carotenoid content in the leaves was significantly higher than those in the other organs, and the stems had higher levels than the inflorescences, green fruits, and black fruits. However, inflorescences (except in M4, M7, and M8) had the lowest total carotenoid contents. The total carotenoid contents in the leaves ranged from 0.57 mg g^−1^ DW (M4) to 0.72 mg g^−1^ DW (M2), and in the inflorescences they ranged from 0.01 mg g^−1^ DW (M1) to 0.12 mg g^−1^ DW (M7). The total carotenoid contents in the leaves were 5.0 to 64.0 times higher than those in the inflorescences (Figure 3B).

### 2.4. Total Ascorbic Acid

The total ascorbic acid content in M6 was the highest (13.09 mg g^−1^ DW) and was significantly higher than those in the other cultivars, while M3 showed the lowest content of ascorbic acid (7.84 mg g^−1^ DW). The organ with the highest content was the inflorescences, and in M2, the content (23.39 mg g^−1^ DW) exceeded those of the other cultivars. Conversely, the lowest content varied among different organs in the different cultivars; it was found in the green fruits in M1 (6.82 mg g^−1^ DW), M2 (6.24 mg g^−1^ DW), M3 (1.15 mg g^−1^ DW), and M8 (4.93 mg g^−1^ DW) and in the stems (3.74–9.03 mg g^−1^ DW) in the remaining cultivars (Figure 4A).

### 2.5. Total Proanthocyanidins, Total Flavonoids, and Total Phenolics

The total proanthocyanidin content in M4 was 17.2 mg g^−1^ DW, which was significantly higher than those in the other cultivars, while M6 (10.98 mg g^−1^ DW) showed a significantly lower total proanthocyanidin content than the other cultivars. The green fruits consistently exhibited the highest total proanthocyanidin content in every cultivar, ranging from 22.21 to 40.29 mg g^−1^ DW, while the stems consistently displayed the lowest content (1.35–2.40 mg g^−1^ DW). With the exception of M6 and M7, the total proanthocyanidin contents in the different organs were ordered from high to low as follows: green fruits, black fruits, inflorescences, leaves, and stems (Figure 4B).

The total flavonoid contents in the eight cultivars ranged from 14.32 mg g^−1^ DW (M4) to 16.45 mg g^−1^ DW (M3). Moreover, the total flavonoid contents in the leaves were higher than those in the inflorescences, except in M3, where the situation was reversed. No significant differences were observed in the total flavonoid contents among the stems, green fruits, and black fruits in most cultivars (Figure 4C).

M6 exhibited significantly more total phenolics than the other cultivars (22.63 mg g^−1^ DW), which was 1.5 times higher than M5 (15.11 mg g^−1^ DW), which had the lowest content. The highest contents were observed in the inflorescences, with the exception of M5 and M7, and the total phenolic contents in the inflorescences ranged from 18.46 mg g^−1^ DW to 32.70 mg g^−1^ DW. However, the lowest contents were observed in the stems (4.74 mg g^−1^ DW–12.32 mg g^−1^ DW). In addition, the total phenolic contents in the inflorescences were 2.6 to 4.9 times higher than those in the stems. The green fruits contained significantly higher total phenolic contents than the black fruits, except in M3 (Figure 4D).

### 2.6. FRAP Analysis

M6 exhibited the highest FRAP level at 0.18 μmol g^−1^ DW, while M5 displayed the lowest at 0.13 μmol g^−1^ DW, and both were significantly different from those of the other cultivars. In addition, the FRAP levels in the green fruits, black fruits, and inflorescences ranged from 0.13 to 0.22 μmol g^−1^ DW, from 0.17 to 0.22 μmol g^−1^ DW, and from 0.14 to 0.20 μmol g^−1^ DW, respectively. Furthermore, the FRAP levels in the leaves were significantly lower than those in the inflorescences but significantly higher than those in the stems, with the stems exhibiting the lowest FRAP levels (0.05 μmol g^−1^ DW–0.12 μmol g^−1^ DW) (Figure 5A).

### 2.7. ABTS Analysis

M6 (42.46%) showed the highest ABTS level, while the lowest ABTS level was found in M3 (34.51%). Meanwhile, the highest ABTS levels were observed in the green fruits (43.39–59.16%), and the ABTS levels in the black fruits were significantly lower than those in the green fruits, while the opposite was observed in M3 and M7. The lowest levels were observed in the stems (except in M3), ranging from 16.84 to 32.72%. In addition, the inflorescences exhibited significantly higher levels than the leaves, except in M3 (Figure 5B).

### 2.8. Principal Component Analysis (PCA)

The first component (PC1) explained 46.4% of the variance, and the second component (PC2) explained 26.6%. In addition, PC1 distinguished three groups: the first group was leaves and stems, the second group was inflorescences, and the last group was green fruits (except in M3). PC2 distinguished two groups: one was leaves and inflorescences, and the other was stems and black fruits (except in M6) (Figure 6A).

The PLS-DA analysis results are shown in Figure 6B. PLS-DA1 explained 46.3% of the variance, and PLS-DA2 explained 26.6%. The distribution was similar to that in the PCA.

According to the results of the loading plot, the major contributors to the leaves were total chlorophyll, total carotenoids, and total flavonoids, while total soluble sugar made a greater contribution to the stems. In addition, the main components in the inflorescences were total soluble protein, total ascorbic acid, and total phenolics. And the major contributor to the green fruits was total proanthocyanidins (Figure 6C).

### 2.9. Correlation Analysis

Two groups of correlations were identified. The first group involved total carotenoids, total flavonoids, and total chlorophyll, while the second group encompassed all other detected substances, excluding total soluble protein (Figure 7). Total carotenoids, total flavonoids, and total chlorophyll were positively correlated with each other, reaching a maximum correlation of 0.98 between total carotenoids and total chlorophyll. In addition, total soluble sugar exhibited negative correlations with total proanthocyanidins, ABTS, and FRAP (with the lowest being -0.83 between total soluble sugar and total proanthocyanidins), while the latter three showed positive relationships. In addition, total phenolics displayed positive relationships with FRAP and total ascorbic acid.

### 2.10. Variance Analysis

The ratios for the cultivar, organ, and interaction (cultivar × organ) variance for all nutritional compounds and antioxidant capacity were significant at the 0.01 level (Table 1).

The variance ratio for the cultivar was highest for total soluble protein (0.499), whereas the highest variances for all other substances and antioxidant capacity were observed in the organ ratio.

## 3. Discussion

In this study, we assessed diverse nutritional components and antioxidant capacity in the inflorescences, green fruits, black fruits, leaves, and stems of eight typical cultivars of *Basella alba*.

The contents of vegetable nutritional components are strongly affected by cultivars [27,28,29]. A study on lettuce highlighted the crucial role of cultivars in determining the types and concentrations of pigments and bioactive components [16]. Ashenaf et al. observed significant differences in chlorophyll contents among three kale cultivars [30]. In addition, variance in antioxidant capacity may exist among different lettuce cultivars [15]. Our findings similarly indicated substantial differences among the eight detected cultivars. Specifically, M5 exhibited the highest total soluble sugar, total soluble protein, and total chlorophyll contents. In addition, M6 displayed the highest total ascorbic acid and total phenolic contents. In contrast, M3 had the lowest content of nutritional components, with the total soluble protein and total ascorbic acid being the lowest. Moreover, M6 demonstrated a higher antioxidant capacity, while the ABTS in M3 was the lowest. These findings hold significance for selecting cultivars based on diverse production requirements in the future.

Substantial variations in nutrient contents among different vegetable organs have been documented [24,31,32]. In comparison to the other organs studied in this study, the inflorescences contained the highest total ascorbic acid and total phenolic contents, which was consistent with previous studies [33,34]. The high total ascorbic acid and total phenolic contents in inflorescences may be linked to the growth and development laws of plants. When a plant transitions from the vegetative growth stage to the reproductive growth stage, the nutritional components in the plant gradually transfer from the vegetative organs to the reproductive organs [35]. In addition, we found relatively elevated levels of total proanthocyanidins in both the green and black fruits. Moreover, compared to the other organs, the measured antioxidant capacities of the green and black fruits were stronger. We could speculate that the high total proanthocyanidin content made a major contribution to the antioxidant capacity, and our correlation analysis results also support this viewpoint. In our previous study, leaves from purple and green flowering stalks exhibited higher pigment, ascorbic acid, proanthocyanidin, flavonoid, and total phenolic contents compared to other edible parts [32]. Similarly, in this study, the total chlorophyll, total carotenoid, and total flavonoid contents were the highest in the leaves of the eight cultivars, with the exception of the total flavonoids in M3. Natalia Drabińska et al. confirmed that to date cauliflower leaves are considered a by-product of cauliflower processing, as they have the highest content of phytochemicals and the highest antioxidant capacity [31]. In addition, cauliflower leaves, which are rich in flavonoids and other compounds, can be used to extend the shelf life of meat products [36], offering innovative ideas for utilizing *Basella alba* leaves. The stems of most cultivars exhibited the highest total soluble sugar contents, which was consistent with prior research [37]. In summary, there are significant differences in the nutrient contents of different organs of *Basella alba*.

Notably, numerous studies have corroborated the presence of phytochemicals in various organs of *Basella alba*, and these chemicals provide various health benefits [14,38]. This may raise concerns about the comprehensive utilization of different organs in chemistry, medicine, and other fields based on the rich nutrients in different organs of *Basella alba*, avoiding the fate of the fruits being discarded and greatly improving the utilization rate of *Basella alba*.

The nutrient contents and antioxidant capacities of plants with different maturities show significant differences. Strawberry fruits undergo changes in soluble solids, total acids, and volatile aroma compounds as they mature, accompanied by a decline in antioxidant capacity [39,40]. In addition, the content of reduced ascorbic acid in the tomato ripening stage is double that in the green stage [41]. Our results demonstrated that green fruits contained higher levels of total chlorophyll, total proanthocyanidins, total phenolics, and ABTS, while black fruits exhibited higher levels of total soluble protein and sugar in most cultivars. The degradation of chlorophyll signals the onset of fruit ripening, aiding in plant growth and responses to environmental changes [42,43]. Previous studies indicated that the capacity to scavenge radicals of an extract, such as total proanthocyanidins (a type of phenolic compound) and total phenolics, decreased significantly with increased maturation, aligning with our results [44,45,46]. This may be explained by the polymerization process of proanthocyanidins during ripening [47]. The molecules of high molecular weight formed through the polymerization process do not easily react with the radical due to their steric hindrance [48], which may be the reason that the antioxidant capacity of the fruits decreased during the ripening process. Meanwhile, the increase in the total soluble protein content may be attributed to a decrease in protease activity [49,50]. García-Gómez et al. found that during the ripening process of apricot (*Prunus armeniaca* L.) fruits, the soluble solid (mainly soluble sugar) content continuously accumulates [51], which is in line with our findings. As discussed, this is due to the fact that fruit ripening is regulated by plant hormones. Increases in sugar substances are accompanied by changes in cell walls and fruit softening [52].

A previous study demonstrated variations in antioxidant capacity results obtained using different measurement methods [53]. Similar patterns emerged in this study, revealing disparities in antioxidant capacity when employing the FRAP and ABTS methods. Specifically, the ABTS method indicated higher antioxidant levels in most of the green fruits compared to the black fruits, but this disparity was not significant when using the FRAP method. A commonly cited explanation for this difference lies in the distinct measurement principles of these two methods. The ABTS method assesses the antioxidants’ capacity to neutralize the 2,20-azinobis (3-ethylbenzthiazolin-6-sulfonic acid) (ABTS^•+^) stable radical cation. However, the FRAP method measures the reduction of a ferric ion (Fe^3+^)–ligand complex to an intensely blue ferrous complex (Fe^2+^) by means of antioxidants in acidic environments [11].

## 4. Materials and Methods

### 4.1. Chemicals and Reagents

Analytical-grade anthrone, ethyl acetate, Coomassie Brilliant Blue G-250, sucrose, phosphoric acid, potassium acetate, quercetin, Folin–Ciocalteu reagent, 300 mmol L^−1^ acetate buffer (pH 3.6 and pH 4.5), 20 mmol L^−1^ ferric chloride, 2,2-azinobis (3-ethyl-benzothiazoline-6-sulfonic acid) (ABTS), and 2.45 mmol L^−1^ ammonium persulfate were obtained from Sangon Biotech Co., Ltd. (Shanghai, China). Analytical-grade sulfuric acid, bovine serum albumin, ethanol, oxalic acid, p-dimethylaminocinnamaldehyde (DMACA), procyanidin B2, aluminum trichloride (AlCl_3_), saturated sodium carbonate, 10 mmol L^−1^ 2,4,6-tripyridyl-S-triazine, 40 mmol L^−1^ HCl, and ferrous sulfate (FeSO4·7H2O) were purchased from Chengdu Kelong Chemical Co., Ltd. (Chengdu, China). Standards of chlorophyll, carotenoids, authentic ascorbic acid, and quercetin were obtained from Solarbio Science & Technology Co., Ltd. (Beijing, China).

### 4.2. Plant Material

Seeds of different cultivars of *Basella alba* were purchased from different breeding companies (Appendix A). Then, eight cultivars of *Basella alba* (M1, M2, M3, M4, M5, M6, M7, and M8) were cultivated at the Wenjiang at Sichuan Agricultural University, Chengdu City, Sichuan Province. In total, 40 healthy plants of each cultivar, which were free of disease or damage, were collected. The 40 plants from each cultivar were divided into four replicate groups. The *Basella alba* plants were washed, dried, and divided into inflorescences, green fruits, black fruits, leaves, and stems. Each part was then mixed for the determination of nutritional components. The samples were then lyophilized in a freeze-dryer and stored at −20 °C until further analysis.

### 4.3. Total Soluble Sugar Content

The lyophilized powder (50 mg) was extracted in 10 mL of distilled water for 20 min at 90 °C, then centrifuged at 4000× *g* for 5 min. Subsequently, a combination of 1 mL of the sample extract, 0.5 mL of an anthrone–ethyl acetate reagent, and 5 mL of concentrated sulfuric acid was added, followed by a 5 min water bath at 90 °C and rapid cooling using ice water. The absorbance values at 630 nm were calculated once the temperature reached room temperature, and the total soluble sugar content was determined using a standard curve of sucrose [20].

### 4.4. Total Soluble Protein Content

The sample (50 mg) was added to 10 mL of distilled water. Then, the solution was stirred for 30 s using a vortex mixer, after which it was stood for 30 min. After that, the solution was centrifuged for 5 min at 4000× *g* and transferred to a polypropylene tube. Then, 1 mL of the supernatant was mixed with Coomassie Brilliant Blue G-250 reagent. The absorbance value was then determined at 595 nm within 20 min after the reaction. The total soluble protein in the samples was calculated based on a standard curve of bovine serum albumin [54].

### 4.5. Total Chlorophyll Content

Frozen powder (50 mg) was ground and extracted with 10 mL of ethanol. After centrifugation at 4000× *g* at room temperature for 5 min, the supernatant was collected, and the total chlorophyll content was measured by measuring the absorbance at 665 nm and 649 nm using a spectrophotometer [55].

### 4.6. Total Carotenoid Content

Frozen powder (50 mg) was extracted with 10 mL of a mixture of acetone and petroleum ether (1:1 *v*/*v*), and the total carotenoid content was determined at 451 nm using a spectrophotometer [55].

### 4.7. Total Ascorbic Acid Content

The sample powder (50 mg) was extracted with 5 mL of 1.0% (*w*/*v*) oxalic acid and centrifuged for 5 min at 4000× *g*. After that, the sample was filtered through a 0.45 μm cellulose acetate filter. Then, 20 μL of the sample was separated on a Waters Spherisorb C18 column (250 × 4.6 mm id; 5 μm particle size) using 0.1% oxalic acid as a solvent at a flow rate of 1.0 mL min^−1^. The total ascorbic acid amount was calculated from the absorbance values at 243 nm using authentic ascorbic acid as a standard. [20].

### 4.8. Total Proanthocyanidin Content

The sample powder (40 mg) was transferred to 4 ml of the extracting reagent (acetone/distilled water/acetic acid = 150:49:1 *v*/*v*). Then, the solution was vigorously vortexed for 5 min, shaken for 1 h, and then centrifuged at 4000× *g* for 5 min. After that, 2.1 mL of p-dimethylaminocinnamaldehyde (DMACA) reagent was added to 700 μL of the supernatant. The total proanthocyanidin content was then determined using spectrophotometry at 640 nm with a standard curve of procyanidin B2 [56].

### 4.9. Total Flavonoid Content

The sample powder (40 mg) was extracted in 50% ethanol. Following 24 h in the dark, the sample solution was centrifuged at 4000× *g* for 5 min. Then, 1.2 mL of the supernatant was combined with 60 μL of 2% aluminum trichloride, 60 μL of 1 mol L^−1^ potassium acetate, and 1.680 mL of distilled water. Absorbance was measured at 415 nm using a spectrophotometer after 40 min, and the total flavonoid content was determined using a standard calibration curve with quercetin in 50% ethanol as a reference standard [20].

### 4.10. Total Phenolic Content

The sample powder (40 mg) was extracted with 50% ethanol. Then, the ethanol extract was centrifuged at 4000× *g* for 5 min, and the supernatant (300 μL) was combined with 1.5 mL of 0.2 mol L^−1^ Folin–Ciocalteu reagent following 24 h of incubation in the dark. Then, 1.2 mL of saturated sodium carbonate was added after 3 min. The absorbance was measured at 760 nm with a spectrophotometer after standing for 20 min at room temperature. Gallic acid was used as a standard [20].

### 4.11. Ferric Reducing Antioxidant Power (FRAP)

The working FRAP reagent was prepared by mixing 300 mmol L^−1^ acetate buffer (pH 3.6), 20 mmol L^−1^ ferric chloride, and 10 mmol L^−1^ 2,4,6-tripyridyl-S-triazine in 40 mmol L^−1^ HCl at a ratio of 10:1:1 (*v*/*v*/*v*). Extracted samples (300 μL) were introduced to a 2.7 mL FRAP working solution, incubated at 37 °C, and vortexed. After 10 min, the absorbance was recorded by a spectrophotometer at 593 nm. The value was calculated based on the FeSO_4_·7H_2_O standard curve, expressed as μmol·g^−1^ of dry weight [57].

### 4.12. 2,2-Azinobis (3-ethyl-Benzothiazoline-6-Sulfonic Acid) (ABTS) Assay

ABTS^+^ radicals were generated by the addition of 2.45 mmol L^−1^ ammonium persulfate to a 7 mM ABTS solution, and the solution was kept in darkness for 16 h. The ABTS^+^ solution was adjusted with acetate buffer (pH 4.5) to an absorbance of 0.700 (±0.020) at 734 nm. For each extracted sample, a 300 μL solution was added to 3 ml of the ABTS^+^ solution. After 2 h, the absorbance was measured spectrophotometrically at 734 nm. The inhibition rate could be calculated as follows [58]:% Inhibition = [(A_control_ − A_sample_)/A_control_] × 100%.

### 4.13. Statistical Analyses

Statistical analysis was performed using the SPSS software package (version 18) (SPSS Inc., Chicago, IL, USA). Data analysis employed a two-way analysis of variance, with the LSD test used for comparisons at a significance level of 0.05. Principal component analysis (PCA) used the unit-variance (UV) scale in SIMCA-P 14.1 (Umetrics, Umeå, Sweden) to explain the relationships between samples. The correlation results were visualized using Cytoscape v. 3.5.1 (The Cytoscape Consortium, New York, NY, USA) [59].

## 5. Conclusions

In this study, we explored the variations in nutritional components and antioxidant capacity in inflorescences, green fruits, black fruits, leaves, and stems of eight typical cultivars of *Basella alba*. Our findings highlight the substantial impact of cultivars on the nutritional composition of *Basella alba*. Notably, M5 and M6 exhibited advantages in both nutrient content and antioxidant capacity, while M3 displayed lower nutritional components. In addition, we observed significant differences in nutrient contents between the different organs of the eight cultivars. Relatively high levels of total proanthocyanidins and antioxidant capacity were found in the green and black fruits, suggesting that proanthocyanidins may contribute significantly to the antioxidant activity in the fruits. Furthermore, our results indicated that the green fruits displayed elevated levels of total chlorophyll, total proanthocyanidins, total phenolics, and ABTS, whereas the black fruits exhibited higher levels of total soluble protein and total soluble sugar in most cultivars. Overall, this research delineates variations in nutritional components and antioxidant capacity among different cultivars, organs, and maturities of *Basella alba*, offering novel insights for the comprehensive utilization of this plant.

## Figures and Tables

**Figure 1 plants-13-00892-f001:**
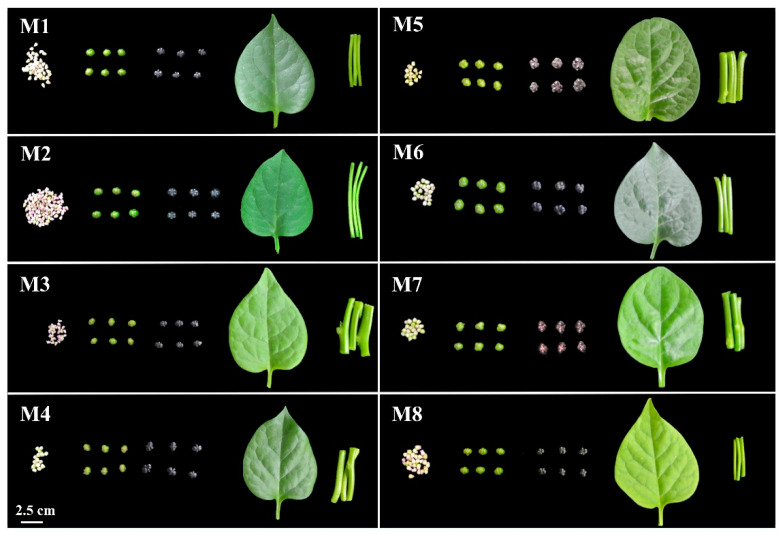
The visual characteristics of inflorescences, green fruits, black fruits, leaves, and stems of eight different cultivars of *Basella alba*.

**Figure 2 plants-13-00892-f002:**
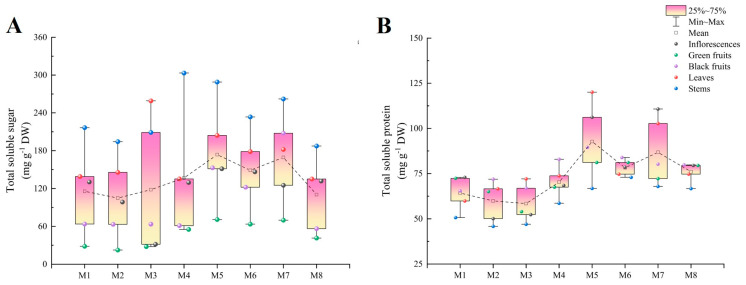
The total soluble sugar (**A**) and total soluble protein (**B**) contents in different organs among eight cultivars of *Basella alba*.

**Figure 3 plants-13-00892-f003:**
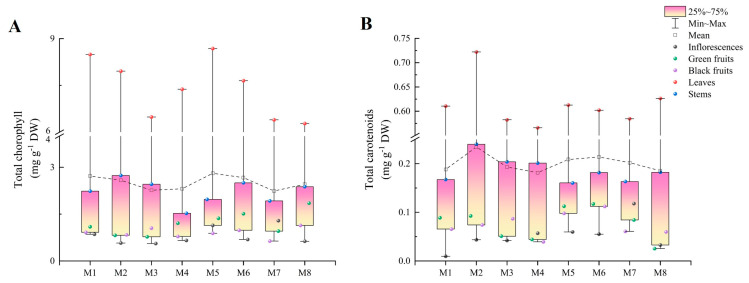
The total chlorophyll (**A**) and total carotenoid (**B**) contents in different organs among eight cultivars of *Basella alba*.

**Figure 4 plants-13-00892-f004:**
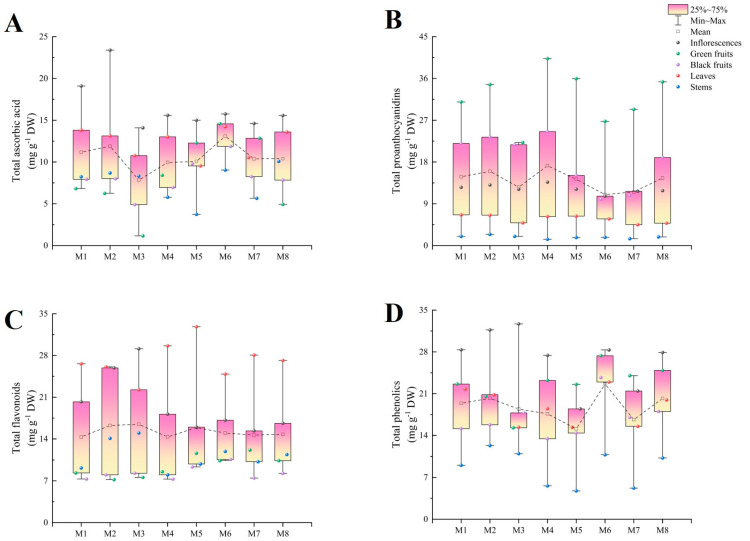
The total ascorbic acid (**A**), total proanthocyanidin (**B**), total flavonoid, (**C**) and total phenolic (**D**) contents in different organs among eight cultivars of *Basella alba*.

**Figure 5 plants-13-00892-f005:**
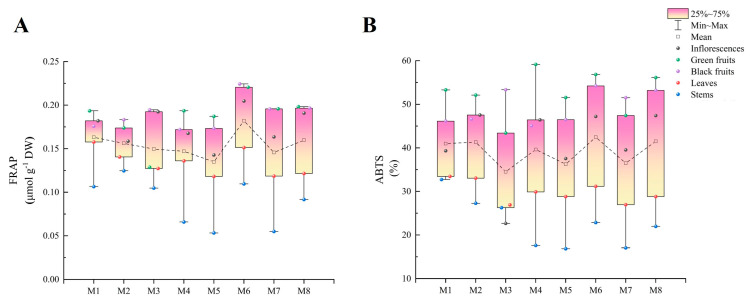
The antioxidant capacities in different organs among eight cultivars of *Basella alba*. (**A**) FRAP and (**B**) ABTS.

**Figure 6 plants-13-00892-f006:**
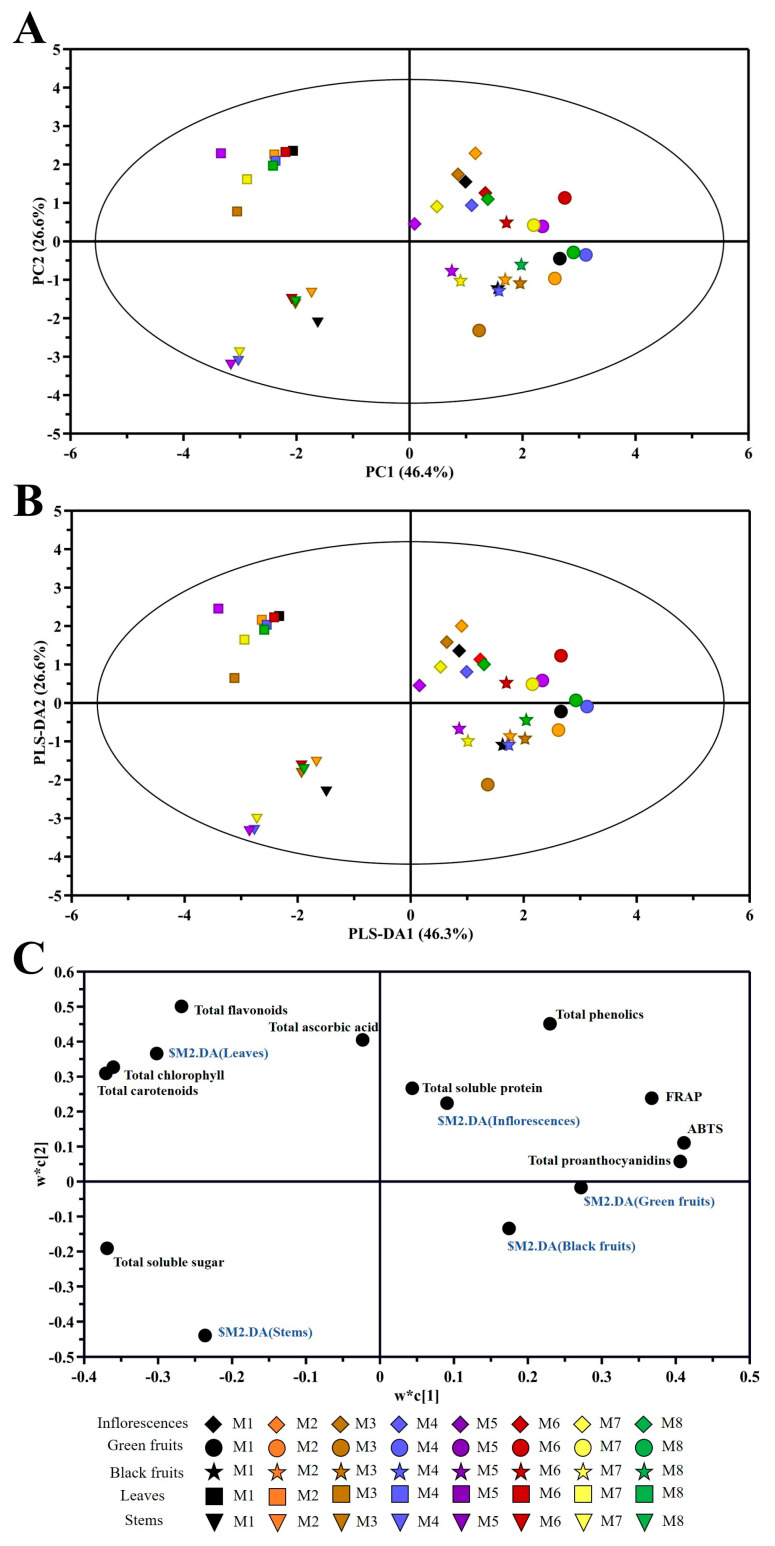
PCA of different organs among eight cultivars of *Basella alba*. (**A**) PCA Score plot; (**B**) PLS-DA Score plot; (**C**) loading plot.

**Figure 7 plants-13-00892-f007:**
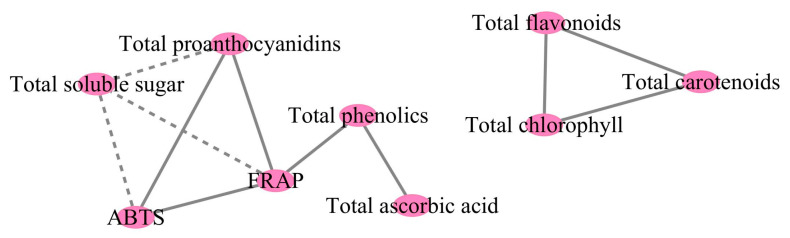
Plot of the correlations between nutritional components and antioxidant capacity in eight cultivars of *Basella alba*. All correlations in the figure reflect absolute values of the Pearson correlation coefficient above the threshold (R2 > 0.65). Solid lines indicate positive correlations, while dashed lines indicate negative correlations.

**Table 1 plants-13-00892-t001:** Estimated proportions of variance components for nutritional components and antioxidant capacity among eight cultivars of *Basella alba*.

Parameter	V_C_/V_P_	V_O_/V_P_	V_CO_/V_P_
Total soluble sugar	0.108 **	0.724 **	0.157 **
Total soluble protein	0.499 **	0.206 **	0.256 **
Total chlorophyll	0.006 **	0.964 **	0.028 **
Total carotenoids	0.006 **	0.961 **	0.015 **
Total ascorbic acid	0.104 **	0.601 **	0.265 **
Total proanthocyanidins	0.033 **	0.901 **	0.062 **
Total flavonoids	0.011 **	0.857 **	0.122 **
Total phenolics	0.097 **	0.759 **	0.129 **
FRAP	0.091 **	0.766 **	0.128 **
ABTS	0.049 **	0.818 **	0.121 **

V_C_/V_P_: ratio of cultivar variance and phenotypic variance; V_O_/V_P_: ratio of organ variance and phenotypic variance; V_CO_/V_P_: ratio of cultivar × organ interaction variance and phenotypic variance. ** indicates significance at the 0.01 probability level in the same column.

## Data Availability

The data presented in this study are available in the manuscript and the Appendix A.

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
