# Peer review of "Variation in Nutritional Components and Antioxidant Capacity of Different Cultivars and Organs of Basella alba"

_plants, 2024, doi:10.3390/plants13060892_

Round 1

Reviewer 1 Report

Comments and Suggestions for Authors

Overall:

·        This paper merits publication in Plants journal

Abstract:

·        The abstract should follow the format of stating the aim or purpose of the study, followed by methodology, results, and conclusion

·        Please list all analyses conducted and specify the corresponding methods used for each

Keywords:

·        Given the similarity between 'antioxidants' and 'antioxidant capacity,' it's advisable to use more specific keywords, such as the names of the antioxidant assays conducted, such as ABTS, FRAP, and TPC. Additionally, to improve the searchability of your paper, consider maximizing the number of keywords allowed by including all the names of the assays tested

Introduction:

·        While the introduction provides a comprehensive overview, it falls short in providing a compelling rationale for why the plant's antioxidant activity was tested. Strengthening this premise would enhance the overall justification for the study.

Results:

·        The results should be presented chronologically in alignment with their description in the methodology. For example, sugars were described first in the methods section before proteins.

·        Based on the methods outlined, where 40 plants were utilized and divided into 4 replicates for each cultivar, could you confirm whether Figures 2-5 depict average values

·        While Figures 2-5 contains informative content, stacking two figures compromises image quality and readability. I recommend preparing individual figures for each image, allowing for clearer presentation and providing each image with its own legend for improved readability.

·        While statistical analysis was performed, it was not clearly presented in the results. I recommend that the authors include this analysis in the results section and, ideally, incorporate it into the figures for enhanced clarity.

·        Section 2.2, line 90, should be: was the stem

·        To improve the results section, consider enhancing the cohesiveness of sentences within paragraphs by incorporating a variety of linking words. This can help strengthen the flow of information and enhance readability

·        Section 2.6. Separate the results for FRAP and ABTS assays

Discussion:

·        While the discussion is substantial, reorganization is needed to chronologically discuss the importance of each analysis and its implications. This approach can improve the clarity and coherence of the section

Materials and methods.

·        Plant Authentication should be included in the methodology

·        I recommend including a separate section detailing all the chemicals and reagents used for the experiments. This would enhance clarity and facilitate the reproducibility of the study.

·        In sections 4.2 to 4.10, the methodologies lack specific information necessary for reproducibility. Please include details such as the mass of the samples, concentrations of reagents and chemicals used, standards and their concentrations, instrumentation, and calculation methods, including the formulas used, and specify if the assays were adapted from existing methodologies. If formulas are included, ensure adherence to the MDPI format for formula presentation to maintain consistency and clarity throughout the paper.

·        In sections 4.2 to 4.8, considering that all these assays measure the total amount in the sample, I recommend including 'Total___ content' for each assay to provide a clearer indication of the total amount measured

·        Section 4.2. followed by a 5-min water bath. (at what temperature?) and what standard was used?

·        Section 4.8 – 4.10. How was the sample extracted?

·         

Comments on the Quality of English Language

Enhancing the choice of vocabulary, employing linking words, and fostering better cohesiveness are essential for refining the text. By selecting more precise terms, incorporating appropriate connectors, and ensuring smoother transitions between ideas, the overall clarity and readability of the content can be significantly improved

Reviewer 2 Report

Comments and Suggestions for Authors

The big group of scientists prepared a manuscript about Basella alba and its cultivars' nutrition value and antioxidants. The aim was to comprehensively understand the utilization value of Basella alba.

The introduction describes the value of the plant, the purpose of the work.

The results are written in detail, but a few notes:

In Figure 2, put the figures next to each other, use the space, and make the legend bigger so that it is clearly readable. Same for other figures. 

The discussion reviews many other studies, although most of them are more than 5 or 10 years old. A little more recent literature, are studies with Basella alba no longer relevant or conducted in the last five years?

Figure 1 is at the beginning of the manuscript but its explanation is at the end in 4.1. Plant material. 

However, the authors performed many biochemical analyzes and extensively studied the composition and nutritional potential of Basella alba.

Round 2

Reviewer 1 Report

Comments and Suggestions for Authors

Dear Authors,

Thank you very much for addressing most of my comments and suggestions. After all, I wanted to see your paper published because it has great scientific soundness and great impact in the field. Adding the necessary information that I thought were important had really improved the readability and  the possible interest of other readers towards your paper.

Here are some of the last bits of tasks that regure your attention prior to acceptance of your paper: 

Figure 2. The second graph needs a label. I guess its B

Figures 2 to 5. The names of the analyses performed were modified in the methodology as total ___ assay, etc. Kindly update the figures with the correct names of the assays (include total ___ assay) and make sure that you update the texts in the manuscript.

Section 4.11. The formulas should be separated from the text and should be formatted based on the MDPI format.

Thank you once again and good luck!

Comments on the Quality of English Language

Check for spelling and grammatical errors.
